Microbiology
Spectrum

# Development and evaluation of the automated multipurpose molecular testing system PCRpack for high-throughput SARS-CoV-2 testing

Yasufumi Matsumura,[1] Tomofumi Nakazaki,[2] Kanako Kitamori,[1,3] Eiki Kure,[1,4] Koh Shinohara,[1] Yasuhiro Tsuchido,[1] Satomi Yukawa,[1] Taro Noguchi,[1] Masaki Yamamoto,[1] Miki Nagao[1]

**ABSTRACT**  Increasing the reliable testing capacity for SARS-CoV-2 is important for the diagnosis and control of COVID-19. We developed an automated, customizable, easy-to-implement molecular testing system, named PCRpack, for the high-throughput testing of SARS-CoV-2. PCRpack includes a liquid handling instrument enclosed in a negative-pressure clean booth (Biomek i5), a laboratory information management system (SimpPCR), other equipment, and documents that are needed for testing operation. An *in vitro* diagnostic assay was employed to detect SARS-CoV-2. System performance was evaluated based on liquid handling accuracy and precision, analytical sensitivity, clinical diagnostic performance, and testing capacity per day. Clinical diagnostic performance was determined against the reference extraction-based reverse transcription-PCR assay using 3,965 upper respiratory samples. Analytical sensitivity analysis showed a lower limit of detection of 1,000 genome copies/mL of sample. The accuracy and precision of sample or reagent dispensing in PCRpack ranged from −2.24% to 0.73% and 0.83% to 4.52%, respectively. In the evaluation of clinical samples, PCRpack showed a positive percent agreement of 96.6% and a negative percent agreement of 100% compared with the reference assay. The average turnaround times per 94 samples and the maximum numbers of tests within an 8-hour shift of one operator were 2 hours and 28 minutes vs 2 hours and 1 minute and 564 vs 376 samples for PCRpack and the manual method, respectively. The developed PCRpack system shows high liquid handling and clinical diagnostic performance and is a promising testing system for increasing the SARS-CoV-2 testing capacity and testing future emerging pathogens.

**IMPORTANCE**  Accurate and fast molecular testing is important for the diagnosis and control of COVID-19. During patient surges in the COVID-19 pandemic, laboratories were challenged by a higher demand for molecular testing under skilled staff shortages. We developed an automated multipurpose molecular testing system, named PCRpack, for the rapid, high-throughput testing of infectious pathogens, including SARS-CoV-2. The system is provided in an all-in-one package, including a liquid handling instrument, a laboratory information management system, and other materials needed for testing operation; is highly customizable; and is easily implemented. PCRpack showed robust liquid handling performance, high clinical diagnostic performance, a shorter turn-around time with minimal hands-on time, and a high testing capacity. These features contribute to the rapid implementation of the high-performance and high-throughput molecular testing environment at any phase of the pandemic caused by SARS-CoV-2 or future emerging pathogens.

**KEYWORDS**  COVID-19, SARS-CoV-2, RT-PCR, automation, liquid handling instrument

Address correspondence to Yasufumi Matsumura, yazblood@kuhp.kyoto-u.ac.jp.

Y.M. received research funds from Beckman Coulter, Nippon Control System, Toyobo, and Precision System Science. M.N. received research funds from Beckman Coulter and Precision System Science. T.N. is an employee of Beckman Coulter, and K.K. is an employee of Nippon Control System. The funding organizations had no role in the study design, data collection and interpretation, or the decision to submit the work for publication.

See the funding table on p. 11.

The COVID-19 pandemic has highlighted the importance of reliable molecular detection tests for the diagnosis and control of the disease (1). Molecular tests such as reverse transcription-PCR (RT-PCR) assays can be developed more rapidly than other types of tests (e.g., antigen tests) and provide highly sensitive and specific results (2). In certain settings, such as hospitals and clinics that manage suspected COVID-19 patients, outbreaks in healthcare-related facilities, or monitoring and screening for specific purposes (e.g., surveillance, mass gathering, quarantine), increasing the testing capacity with a short turn-around time is often needed. Japan has experienced several COVID-19 waves (3), and there has been both a continuous demand and surges in demand for molecular tests. The average number of people who underwent RT-PCR tests per day before the first wave (April 2020) was 569, but it expanded to 19,923 during the third wave (August 2020) and 161,992 in 2022, according to the Ministry of Health, Labour and Welfare of Japan (https://www.mhlw.go.jp/stf/covid-19/kokunainohasseijou-kyou_00006.html). An up to fivefold increase in the number of tests was observed in the sixth wave that started in January 2022.

At the same time, in this COVID-19 pandemic, we have been challenged by global supply shortages and the need for skilled laboratory professionals (4, 5). Molecular tests, especially laboratory-developed or in-house assays (e.g., the World Health Organization-accredited Corman's assay) (6), require specialized skills (7). Adding new staff may be difficult because hiring and training staff members while responding to the high demand for tests requires substantial effort, and training for molecular diagnostics involves a robust education curriculum (8, 9).

In these situations, an accurate, easy-to-implement testing system that can process hundreds of samples with a minimal workload while assuring testing quality is needed. This motivated us to develop an automated multipurpose molecular testing system for the rapid, high-throughput testing of infectious pathogens, named PCRpack. PCRpack was designed to fulfill the above feature criteria and included everything needed to initiate the molecular diagnostic testing of SARS-CoV-2. PCRpack includes a customized liquid handling instrument (Biomek i5, Beckman Coulter, Tokyo, Japan), a laboratory information management system (LIS) (SimpPCR, Nippon Control System, Yokohama, Japan), a real-time PCR instrument, other equipment needed for the molecular detection of SARS-CoV-2, and documents or templates for testing and laboratory management (e.g., standard operating procedure; Fig. 1). PCRpack employs an *in vitro* diagnostic assay to detect SARS-CoV-2. In this study, we aimed to evaluate the PCRpack system and determine its liquid handling performance, analytical sensitivity, clinical diagnostic performance, and testing capacity for the molecular detection of SARS-CoV-2.

## RESULTS

### Liquid handling performance

PCRpack used a TRexGene SARS-CoV-2 detection kit (Toyobo, Osaka, Japan) for direct RT-PCR testing, which bypasses a nucleic acid purification step. The assay requires the dispensing of 3-, 8-, and 40-µL volumes (Fig. 2). The accuracy and precision of the PCRpack system ranged from −2.24% to 0.73% and from 0.83% to 4.52%, respectively (Table 1). These values were superior to those of manual handling in 3-µL reagent dispensing but were inferior in 8-µL sample and 40-µL reagent dispensing.

### Analytical sensitivity

The limit of detection (LOD) of the PCRpack system was determined to be 1,000 copies/mL for both saliva and nasal swabs (Table S1).

### Clinical diagnostic performance

A total of 3,965 respiratory samples (saliva, $n$ = 2,198; nasopharyngeal swabs, $n$ = 1,177; nasal swabs, $n$ = 590) were included in the analysis. For all specimen types, the test results of the PCRpack system were highly concordant with the reference assay (kappa

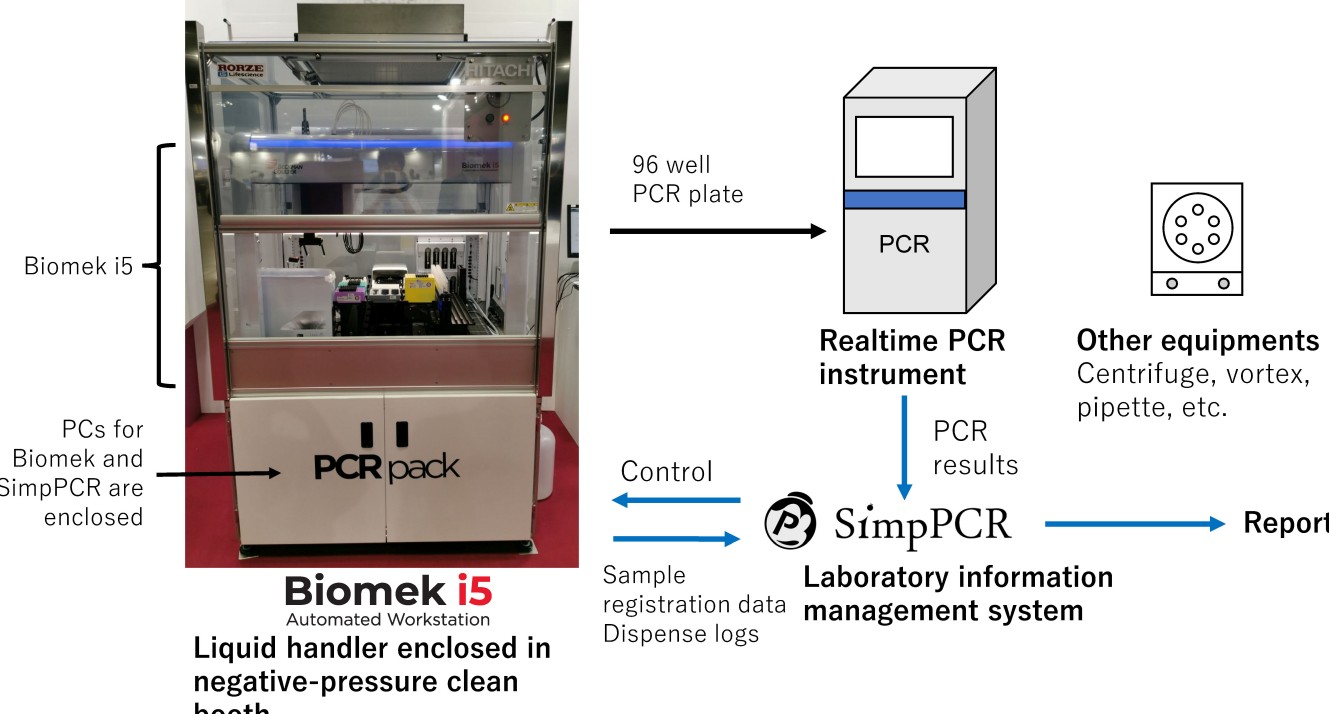

**FIG 1** Components of the PCRpack system. The negative pressure clean booth has casters and is movable. The dimensions of the system are a height of 2,063 mm, width of 1,256 mm, and depth of 990 mm. The system requires two independent power supplies of 100 V and 15 A: one for the clean booth, including the liquid handler and laboratory information management system, and the other for the PCR instrument.

values of 0.97–0.99; Table 2). The positive percent agreement ranged from 95.6% to 99.0% according to specimen types, and the negative percent agreement was 100% in all specimen types. No samples were classified as invalid, including 15 discordant samples (saliva, $n = 3$; nasopharyngeal swab, $n = 11$; nasal swab, $n = 1$) that showed negative results by PCRpack (with a positive internal control) and positive results by the reference assay. These samples were tested using the Xpert Xpress SARS-CoV-2 (Beckman Coulter) assay, and all 15 samples were determined to be positive. The cycle threshold (Ct) values of these discordant samples were high when compared to the other reference assay-positive samples (mean, 33.9 vs 24.7; Fig. S2).

## Test time and cost

Compared to the manual method, the average turnaround time of the PCRpack system was approximately 30 minutes shorter (2 hours and 28 minutes vs 2 hours and 1 minute; Fig. 3), and the average hands-on time was approximately 1 hour shorter (1 hour and 23 minutes vs 25 minutes). The maximum numbers of samples that could be tested within an 8-hour-day shift of one operator were 564 samples (six batches) for PCRpack and 376 samples (four batches) for the manual method (Fig. S3). With 94 samples per batch, the cost for one sample was $12.6 for PCRpack and $12.3 for the manual method: $12.5 and $11.9 for consumables and $64 (2 hours and 33 minutes hands-on time per 564 samples) and $140 (5 hours and 35 minutes per 376 samples) for labor, respectively (Dataset S3).

## DISCUSSION

### PCRpack testing system features

The PCRpack system has several unique features that are different from other high-throughput automated molecular testing platforms (10), including a customizable liquid

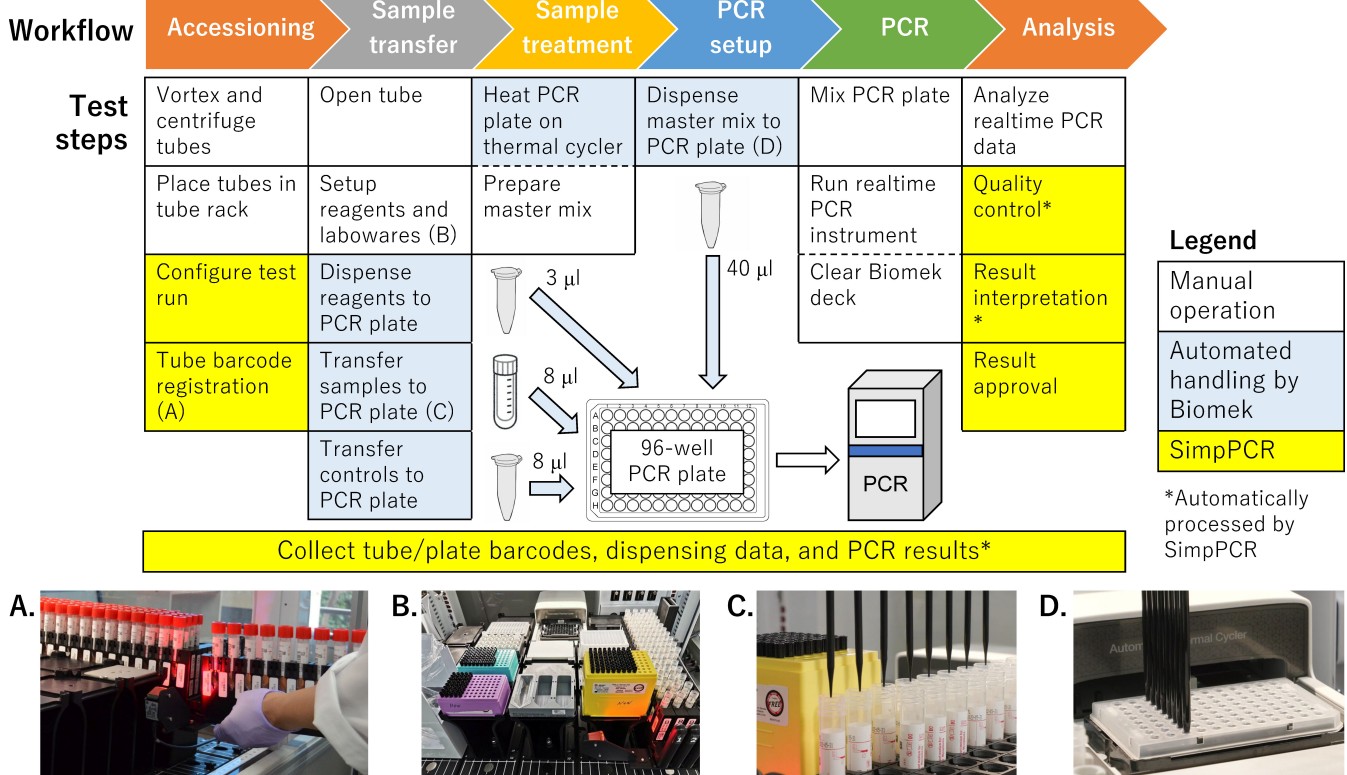

**FIG 2** The workflow of the PCRpack system. The workflow was divided into six procedures, and each procedure included several test steps. A broken line between steps indicates that the below and above steps were performed simultaneously. Panels A to D are provided to explain the steps indicated by the characters within parentheses. In the "transfer samples to PCR plate" step (panel C), a maximum of 94 samples were transferred from specimen tubes to a temporary deep-well plate and then transferred from the deep-well plate to a 96-well PCR plate. After dispensing controls into the 96-well PCR plate, the system can be paused to perform a visual inspection of the 96-well PCR plate to verify the dispensing of reagents and samples.

handling instrument within a negative pressure booth, integration of LIS, and inclusion of equipment and documents for operation. Strategies for increasing testing capacity include partial/full automation of the testing process, direct RT-PCR assay without an RNA extraction procedure, and pooled testing (11). We adopted the former two approaches because pooled testing requires a longer turn-around time at least for all positive samples and involves a potential contamination risk. Laboratory automation can contribute to reducing errors and improving quality while saving labor and costs (12). PCRpack can be easily customized. In the SimpPCR operating window, the specimen types and transfer volumes of reagents and/or samples can be modified. Changes in testing protocols (number of reagents, editing of testing steps) are also possible upon request, but they may require validation studies by a skilled laboratory staff according to the level of customization. Even in the early stage of an emerging infectious disease when *in vitro* diagnostics are not available or there is a severe shortage of reagents

**TABLE 1** Liquid handling performance of the PCRpack system for the detection of SARS-CoV-2[a]

| Material | Intended volume, µL/well | PCRpack | | | Manual | | |
|---|---|---|---|---|---|---|---|
| | | Average volume, µL/well (range, SD) | Accuracy | Precision, CV | Average volume, µL/well (range, SD) | Accuracy | Precision, CV |
| Pretreatment reagent | 3 | 3.03 (2.99–3.05, 0.03) | 0.73% | 0.83% | 2.81 (2.63–2.91, 0.13) | 6.48% | 4.68% |
| Sample[b] | 8 | 7.82 (7.29–8.04, 0.35) | 2.24% | 4.52% | 8.45 (8.31–8.59, 0.10) | 5.65% | 1.19% |
| RT-PCR master mix | 40 | 39.24 (38.74–40.33, 0.64) | 1.90% | 1.62% | 40.15 (39.79–40.44, 0.24) | 0.37% | 0.60% |

[a]SD, standard deviation; CV, coefficient of variation.
[b]Clinical saliva samples were collected using a 1 mL cotton swab, which was submerged in 3 mL viral transport media.

**TABLE 2** Clinical diagnostic performance of the PCRpack system for the detection of SARS-CoV-2[a]

| Specimen | Number of positive/negative samples tested | Positive percent agreement (95% CI) | Negative percent agreement (95% CI) | Kappa (95% CI) |
|---|---|---|---|---|
| Saliva | 93/2,105 | 99.0% (96.9–100%) | 100% (99.8–100%) | 0.98 (0.96–1) |
| Nasopharyngeal swab | 247/930 | 95.6% (93.0–98.1%) | 100% (99.6–100%) | 0.97 (0.95–0.99) |
| Nasal swab | 96/494 | 96.8% (93.2–100%) | 100% (99.3–100%) | 0.99 (0.98–1) |
| All | 436/3,529 | 96.6% (94.4–98.1%) | 100% (99.9–100%) | 0.98 (0.97–0.99) |

[a]CI, confidence interval.

and/or labware, as we experienced in the COVID-19 pandemic (4, 5), PCRpack can be modified to use in-house assays or other available reagents/labware. This feature also contributes to a low running cost of approximately $13 per sample. The cost can even be lowered to approximately $3 when an in-house reagent is used. Due to biosafety considerations during the liquid handling of infectious samples (13), most fully automated testing systems require viral inactivation and/or sample transfer within a biological safety cabinet before sample loading, which leads to increased hands-on time (14, 15). PCRpack has a liquid handling instrument enclosed in a negative pressure booth so that raw samples can be directly loaded and the entire liquid handling process, including heat inactivation, is performed without interruption. This contributes to operator biosafety and reduces hands-on time.

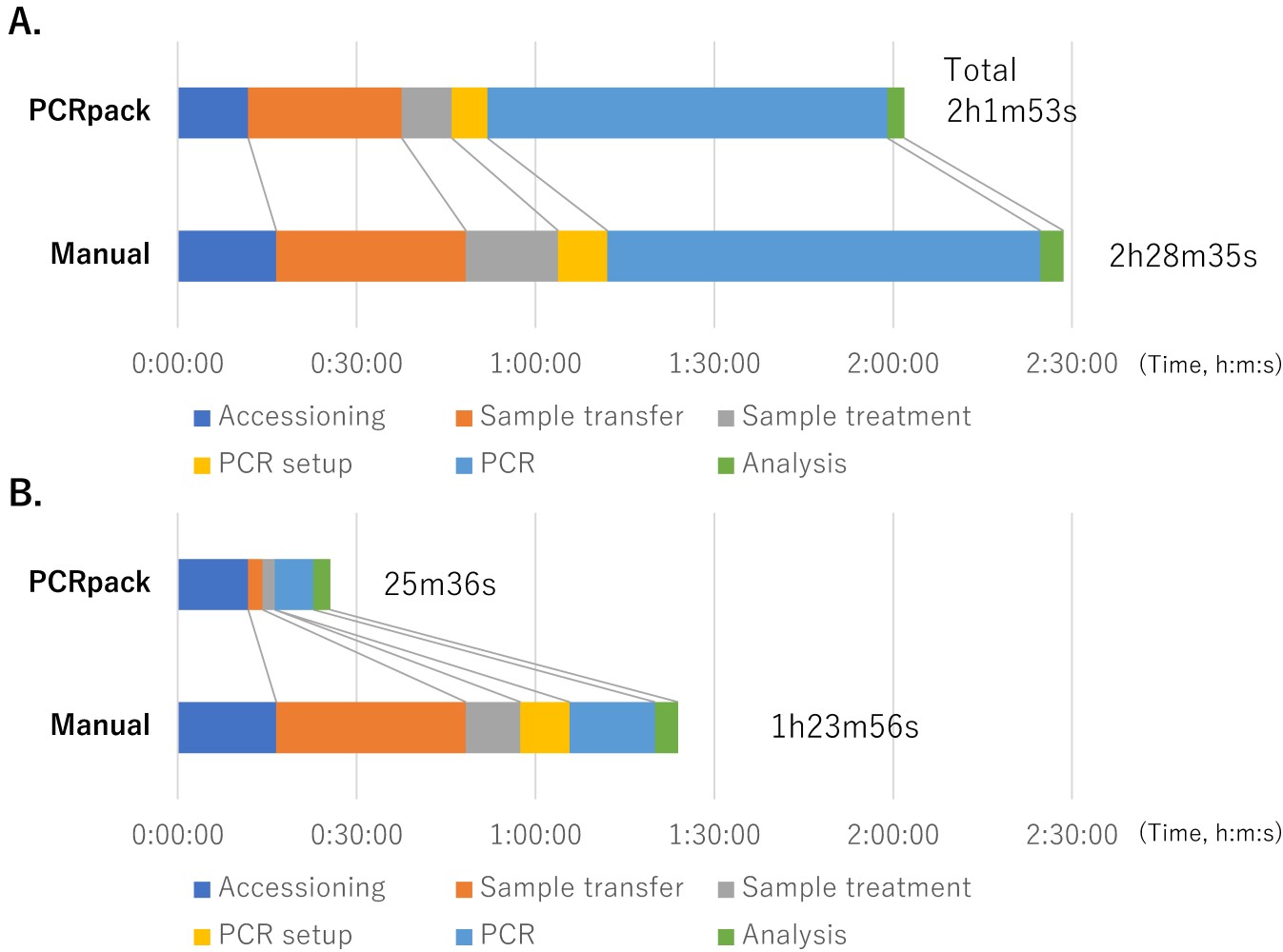

**FIG 3** Comparison of test times for each test step between the PCRpack and manual testing methods. Panel A shows the total elapsed time, and panel B shows the hands-on time. The raw data are available in Dataset S2.

The LIS of PCRpack, SimpPCR, can integrate different instrumentation and provide accurate sample tracking, quality control, result interpretation, and reporting, contributing to accurate testing and shortening turn-around times. During surges of molecular testing in the COVID-19 pandemic, US public health laboratories were challenged by unceasing demand for faster accurate testing under skilled staff shortages (9). Based on these lessons, the authors proposed the introduction of LIS and automation for high-throughput testing and reducing errors from manual operations (9). SimpPCR can either be run alone or be placed downstream of another LIS. This feature, with other materials needed for SARS-CoV-2 testing, enables easy and rapid implementation of PCRpack in laboratories even without adequate experience with molecular diagnostics.

The operation of PCRpack is easier than the manual method due to the automation of testing steps. In our laboratory, less time was needed for staff training for PCRpack than for the manual method (3 days vs 10 days). This feature, combined with the shorter hands-on time, can be a strength of PCRpack in cases of skilled staff shortages.

The size of the PCRpack system is compliant with a large size, high-height van (Fig. 1). Future studies are needed to optimize and validate a mobile PCRpack laboratory. The need for two independent power supplies may restrict its utility in emergency field centers with an inadequate or unstable power supply.

## Liquid handling performance

Accurate liquid handling is vital for molecular testing. According to the Biomek i Series Operational Qualification Manual (Beckman Coulter), the pipetting accuracy of the Biomek liquid handler is qualified by confirming an accuracy of ±3% and precision of <5% in dispensing a 10-µL volume. After the optimization of pipetting parameters, another liquid handling instrument, TECAN Freedom EVO, showed accuracy/precision values of 6.6%/10.2% and 2.3%/4.9% in dispensing 3-µL and 9-µL phosphate-buffered saline, respectively (16). The PCRpack system fulfilled the above Biomek criteria for the three dispensing volumes required for the assay, although the reagents and sample (viral transport media) were viscous (Table 1). The accuracy of manual handling exceeded these values in dispensing pretreatment reagents and samples (−6.48% and 5.65%, respectively). This might be related to the superiority of the liquid handler in dispensing small volumes of viscous liquids. In contrast, the precision of sample dispensing by PCRpack was inferior to that of manual handling (4.52% vs 1.19%). To reduce the number of tips used while avoiding the introduction of air bubbles in the sample transfer step involving the PCR volume, PCRpack used the same tip to transfer the samples from specimen tubes (with 3-mL liquid) to deep-well plates and from the deep-well plates to PCR plates. The remnants from the first dispensing step might influence the volume of the second dispensing step. It is noted that these evaluations were limited in terms of the use of a gravimetric method and calculation of the dispensed volume per well by using an average of 96 wells, which could estimate precision better than the actual values.

## Analytical sensitivity

We employed TRexGene as a molecular assay because it can be performed without a nucleic acid purification step while showing high sensitivity and specificity, contributing to rapid and accurate testing (17). The defined LODs of the PCRpack system (1,000 copies/mL) were the same as the results from our previous study performed with a SARS-CoV-2 Detection Kit -Multi- (the former product name of TRexGene) by manual handling (17). This indicates that automation by the PCRpack system does not compromise analytical sensitivity.

## Clinical diagnostic performance

In our previous study, the TRexGene assay showed 100% positive percent agreement and 100% negative percent agreement for both nasopharyngeal swabs and saliva samples when compared with the NIID N2 reference assay (17). Similarly, automation by the

PCRpack system achieved 96.6% positive percent agreement and 100% negative percent agreement with the NIID N2 reference assay. Discordant samples showed high Ct values by the reference assay, which showed a lower LOD (391 copies/mL) (18) than the TRexGene assay. These samples might not be detectable by the PCRpack system due to their low viral loads below the LOD of TRexGene and above that of the reference assay. A perfect negative percent agreement indicates the absence of on-deck contamination. Liquid handlers have a potential risk of errors with viscous samples (such as saliva), but there were no differences in the diagnostic performance among the different sample types (Table 2). To avoid dispensing problems, we utilized a swab-based saliva sampling method that traps mucous secretions and centrifuged all samples to avoid dispensing mucus-heavy portions. These results indicate that the clinical diagnostic performance of the PCRpack system was defined by the TRexGene assay and that the automated liquid handling of the PCRpack system was accurate and reliable. In addition, the automated liquid handling and LIS of the PCRpack system reduced both the turn-around time and hands-on time (Fig. 3), enhancing the maximum number of test samples assessed per day by 1.5 times (Fig. S3).

## Test time and cost

Several high-throughput fully automated molecular testing platforms for the detection of SARS-CoV-2, such as Roche Cobas 6800, Hologic Panther, Abbott Alinity m, and NeuMoDx 288, have been validated to show comparable clinical diagnostic performance with comparators (11). These systems were reported to have testing capacities of 220 to 288 samples during an 8-hour shift, with 50–90 minutes of hands-on time and 1 hour and 20 minutes–4 hours and 18 minutes of turn-around time (14, 19, 20). The Roche Cobas 8800 and Hologic Panther Fusion systems have higher capacities, but detailed data in real-life settings are lacking for these systems. In contrast to these systems, although PCRpack is not fully automated, it shows a comparable or superior testing capacity, hands-on time, or turn-around time (Fig. 3). The testing capacity of PCRpack is limited by the PCR step rather than the liquid handling step (Fig. S3). Adding another PCR instrument can enhance the testing capacity to 752 samples (eight batches) within an 8-hour shift (Fig. S4).

The consumables cost of PCRpack may be lower than that of other fully automated platforms (approximately $13 vs $40–$60 in Japan). Compared with the manual method, PCRpack requires an additional running cost of $0.3/sample plus initial and maintenance costs. The maintenance of PCRpack does not require component replacement at regular intervals, but an annual maintenance inspection is recommended. The cost-effectiveness and merits of PCRpack depend on individual situations, including requirements for testing capacity and availability of skilled staff. A larger number of tests may increase the risk of error in the manual method, giving the accurate automated testing by PCRpack an advantage.

## Limitations

This study has several limitations. First, liquid handling performance and test time measurements were performed by only one operator, and differences and/or variabilities among operators could not be evaluated. Second, the study lacked validation by multiple investigators or in multiple locations, different molecular assays, and samples obtained from different clinical backgrounds. This lack might limit the generalizability of the findings to other laboratories, other assays or pathogens, and other patient populations. Third, we could not assess the clinical significance of false-negative samples due to a lack of clinical information. Fourth, while PCRpack was designed to reduce hands-on time and operator interventions, it still relies on skilled operators for its proper functioning. Operator expertise, training, and experience can influence the system's performance. Fifth, we could not perform a comprehensive analysis including initial investment, maintenance, and operational costs over time. These costs could

vary in different settings and regions. The strengths included the use of a relatively large number of samples for different specimen types and the coevaluation of system performance (liquid handling performance, turn-around time, and task load), in addition to clinical testing performance.

## Conclusions

The developed PCRpack automated molecular testing system showed robust liquid handling performance, high clinical diagnostic performance, a shorter turn-around time with minimal hands-on time, and a high testing capacity, with one operator for the molecular detection of SARS-CoV-2. The system is provided in an all-in-one package, including everything needed for testing, is highly customizable, and is easily implemented. These features contribute to the rapid implementation of the high-performance and high-throughput molecular testing environment at any phase of the pandemic. PCRpack is a promising molecular testing system for clinical laboratories, centralized laboratories, or research laboratories that process hundreds of samples for the detection of SARS-CoV-2 and those that need to be prepared for other pathogens, including the next emerging pathogen that will cause a pandemic.

## MATERIALS AND METHODS

### Development of PCRpack

A liquid handling instrument (Biomek i5 Span-8 without enclosure containing eight independent pipette channels, Beckman Coulter), a barcode reader (Tube Scan, Beckman Coulter), and an on-deck thermal cycler (ATC, Thermo Fisher Scientific, Waltham, MA, USA) were employed to execute all automated workflows from sample transfer to RT-PCR setup (the steps colored cyan in Fig. 2) and was controlled by SimpPCR. Liquid-level sensing with conductive tips was enabled to ensure aspiration and dispensing of the intended volume. The instrument was enclosed in a negative pressure clean booth (RLS Negative Pressure Clean Booth for Biomek i5, Beckman Coulter) with inlet and outlet fan filter units using a HEPA filter that achieved ISO1644-1 Class 6 cleanliness and −10-Pa pressure. Airflow visualization using a water mist device (PROFECIO Avis, Goldwin, Tokyo, Japan) confirmed the absence of outward airflow even when the front door opened at a maximum height of 400 mm.

We developed a web-based LIS for molecular testing in clinical laboratories named SimpPCR. SimpPCR controls and collects logs from Biomek and Ct values from the real-time PCR instrument and automatically determines test results according to the interpretation criteria (the steps colored yellow in Fig. 2). After the approval of the results by an operator, test reports can be automatically generated. PCRpack is available from Beckman Coulter.

A PCRpack operator needs basic laboratory skills (pipetting, centrifuge, and vortex). The 3-day training program for a technical assistant includes proficiency in the standard operating procedure; operation of the instruments (liquid handler and PCR instrument) and LIS; and knowledge of biosafety level 2, assay interpretation, reporting, and quality assurance.

### RT-PCR assay for PCRpack

An *in vitro* diagnostics TRexGene SARS-CoV-2 Detection Kit was used for direct RT-PCR testing according to the manufacturer's instructions. First, 8 µL of raw samples was mixed with 3 µL of sample treatment reagents, followed by heating at 95℃ for 5 minutes and the addition of 40 µL of RT-PCR reagents. Fluorescent signals of the SARS-CoV-2 N1 (Cy5 dye) and N2 (ROX) targets and the internal control (FAM) were detected on the QIAquant 96 5plex (Qiagen, Hilden, Germany) real-time PCR instrument. Thermal cycling was performed as follows: 5 minutes at 42℃ and 10 seconds at 95℃, followed by 45 cycles of 5 seconds at 95℃ and 30 seconds at 60℃. The Ct values were calculated by

using the automatic determination of the fluorescence threshold function. The results were interpreted as positive when the Ct values were ≤40 for either the N1 or N2 genes and negative when the Ct values were >40 for both N1 and N2 genes and the Ct value of the internal control was ≤40. All other results were regarded as invalid. For quality control, positive and negative controls from the AccuPlex SARS-CoV-2 Reference Material Kit (SeraCare, Milford, MA, USA) were used for each batch.

## Liquid handling performance

The performance of PCRpack vs manual handling was compared according to accuracy (the difference between the actual volume dispensed and the intended volume, divided by the intended volume) and precision (coefficient of variation, CV, calculated by the standard deviation divided by the mean). A total of 94 clinical saliva samples and two controls were tested using a 96-well PCR plate (MicroAmp EnduraPlate Optical 96-Well Blue Reaction Plates with Barcode, Thermo Fisher Scientific). Following the standard operating workflows of PCRpack and the manual method, 450-µL clinical samples were initially aliquoted into a 96-well deep-well plate, and 8-µL aliquots were then dispensed from the deep-well plate into the PCR plate. The actual dispensed volume per well was calculated using the gravimetric method (i.e., the difference in the weight of the plate before and after dispensing divided by the density). The densities of the reagents were determined by an average weight per 1 cm$^3$ that was determined by triplicate measurements. The measurement was performed five times by the same skilled operator on different days. Eight-channel micropipettes (Pipet-Lite XLS LTS; Mettler-Toledo Rainin; Oakland, CA, USA) were used for manual dispensing. All experiments were performed in an air-conditioned room at 25℃.

## Analytical sensitivity

We determined the LOD of the PCRpack system using 20 replicates of serial dilutions of the heat-inactivated SARS-CoV-2 strain (ATCC VR-1986HK) at concentrations of 500, 1,000, and 2,500 genome copies/mL in a negative matrix (pooled saliva and nasal swab samples that tested negative by the reference assay). The LOD was defined as the lowest concentration at which 19 of 20 (95%) replicates were positive.

## Clinical specimens

Between August 2020 and March 2022, respiratory samples in viral transport media (UTM; Copan, Brescia, Italy) were submitted to the laboratory at the Kyoto University Hospital for SARS-CoV-2 RT-PCR testing because of clinical suspicion of COVID-19 or for screening of contacts of COVID-19 clusters. The Kyoto University Hospital is a 1,141-bed tertiary academic center located in Kyoto, Japan, which has a population of ≈1 million. Saliva was collected by placing a proprietary swab (Sysmex, Kobe, Japan) that can absorb 1 mL of saliva in the mouth without rubbing for at least 1 minute in compliance with the manufacturer's instructions. Test examinees were asked not to eat or drink 30 minutes before the collection of saliva. Flocked swabs (Copan) were used for the collection of anterior nasal and nasopharyngeal specimens. For nasal swab collection, a swab was inserted into the first nostril until the swab tip is no longer visible and rotated against the wall of the nostril in a large circular path five times. The same swab was used for the specimen collection of the other nostril, and the same procedure was repeated. For nasopharyngeal swab collection, a swab was into the nostril, parallel to the palate until the swab reached a depth equal to the distance from the nostrils to the outer opening of the ear or the examiner detected resistance. The swab was left in place for 10 seconds to absorb secretions and then it was removed slowly while rotating it. After collection procedures for saliva, nasal, and nasopharyngeal swabs, they were submerged in 3 mL of viral transport media, and their shafts were broken to leave the swabs in the media. The samples that showed an adequate remaining volume for the study were eligible and were prospectively stored at −80℃ until tested by PCRpack and the

reference assay. Therefore, the clinical diagnostic performance study was not influenced by overburdened settings during surges for tests.

## Reference RT-PCR assay

The N2 assay developed by the National Institute of Infectious Disease (NIID) in Japan (21) was used as the reference standard. RNA was extracted from 200-µL respiratory samples using the MagNA Pure 96 DNA and Viral NA Small Volume extraction kit and a MagNA Pure 96 Instrument (Roche, Basel, Switzerland) and was eluted in a final volume of 50 µL. RT-PCR was performed using 5 µL of extracted RNA, 0.5 pM of NIID_2019-nCOV_N_F2 forward primer (AAATTTTGGGGACCAGGAAC), 0.7 pM of NIID_2019-nCOV_N_R2 reverse primer (TGGCAGCTGTGTAGGTCAAC), 0.2 pM of NIID_2019-nCOV_N_P2 probe (FAM-ATGTCGCGCATTGGCATGGA-BHQ1), and TaqPath 1-Step RT-qPCR Master Mix, CG (Thermo Fisher Scientific) in a 20-µL reaction volume on a LightCycler 480 System II (Roche, Basel, Switzerland). Thermal cycling was performed as follows: 2 minutes at 25°C, 15 minutes at 50°C, and 2 minutes at 95°C, followed by 45 cycles of 3 seconds at 95°C and 30 seconds at 60°C. The Ct values were determined by the second derivative maximum method. The results were interpreted as positive when the Ct value of the N2 gene was <40 and as negative when it was ≥40. The samples for which the results were discordant between PCRpack and reference assay were tested using the *in vitro* diagnostics Xpert Xpress SARS-CoV-2 assay according to the manufacturer's instructions.

## Test time measurement and cost estimate

The time needed to perform each test step for the testing of 94 samples was measured in triplicate for the PCRpack and manual methods. Trials were performed by the same skilled operator on different days. In the manual method, the same RT-PCR testing performed by the PCRpack system was performed without the Biomek i5 instrument, but SimpPCR was used for sample management and RT-PCR result interpretation (Fig. S1). The turnaround time and timeline of each batch within an 8-hour-day shift were calculated. Running costs for the PCRpack and manual methods were calculated from the costs for consumables (reagents and plasticware) and labor (the total hands-on time within an 8-hour shift multiplied by labor costs per hour ($25). The costs were estimated at an exchange rate of $1 = 120 yen.

## Statistical analysis

The agreement between the PCRpack and reference assays was assessed by Cohen's kappa concordance coefficient. All statistical analyses were performed using SAS Studio 3.8 (SAS Institute Inc., Cary, NC). Visualization of the Ct values was conducted using R (https://cran.r-project.org) and ggplot2 (https://ggplot2.tidyverse.org).

## ACKNOWLEDGMENTS

This research was supported by the COVID-19 Private Fund (to the Shinya Yamanaka laboratory, CiRA, Kyoto University), Nippon Control System, and Beckman Coulter.

Yasufumi Matsumura: conceptualization, data curation, formal analysis, funding acquisition, investigation, methodology, project administration, resources, supervision, validation, visualization, writing—original draft, writing—review and editing. Tomofumi Nakazaki: conceptualization, data curation, funding acquisition, methodology, resources, investigation, software, validation, writing—review and editing. Kanako Kitamori: funding acquisition, methodology, investigation, software, writing—review and editing. Eiki Kure: conceptualization, investigation, validation, writing—review and editing. Koh Shinohara: investigation, writing—review and editing. Yasuhiro Tsuchido: investigation, writing—review and editing. Satomi Yukawa: investigation, writing—review and editing. Taro Noguchi: investigation, writing—review and editing. Masaki Yamamoto: investi-

gation, data curation, resources, writing—review and editing. Miki Nagao: funding acquisition, resources, supervision, writing—review and editing.

## AUTHOR AFFILIATIONS

[1]Department of Clinical Laboratory Medicine, Kyoto University Graduate School of Medicine, Kyoto, Japan
[2]Beckman Coulter, Tokyo, Japan
[3]Nippon Control System, Yokohama, Japan
[4]Faculty of Pharmacy, Kyoto Pharmaceutical University, Kyoto, Japan

## AUTHOR ORCIDs

Yasufumi Matsumura  http://orcid.org/0000-0001-8595-8944

## FUNDING

| Funder | Grant(s) | Author(s) |
| --- | --- | --- |
| COVID-19 Private Fund (to the Shinya Yamanaka laboratory, CiRA, Kytoo, Japan) | | Yasufumi Matsumura |
| | | Masaki Yamamoto |
| | | Miki Nagao |
| Nippon Control System | | Yasufumi Matsumura |
| Beckman Coulter | | Yasufumi Matsumura |

## AUTHOR CONTRIBUTIONS

Yasufumi Matsumura, Conceptualization, Data curation, Formal analysis, Funding acquisition, Investigation, Methodology, Project administration, Resources, Validation, Visualization, Writing – original draft, Writing – review and editing | Tomofumi Nakazaki, Conceptualization, Investigation, Software, Validation, Writing – review and editing | Kanako Kitamori, Conceptualization, Software, Validation, Writing – review and editing | Eiki Kure, Investigation, Resources, Writing – review and editing | Koh Shinohara, Resources, Writing – review and editing | Yasuhiro Tsuchido, Resources, Writing – review and editing | Satomi Yukawa, Resources, Writing – review and editing | Taro Noguchi, Resources, Writing – review and editing | Masaki Yamamoto, Resources, Writing – review and editing | Miki Nagao, Funding acquisition, Resources, supervision, Writing – review and editing

## DATA AVAILABILITY

All data used in this study are available within the manuscript and supplemental materials.

## ETHICS APPROVAL

This study was performed in line with the principles of the Declaration of Helsinki. The Ethics Committee of Kyoto University Graduate School and the Faculty of Medicine approved this study (R2379), and the need to obtain informed consent from each patient was waived.

## ADDITIONAL FILES

The following material is available online.

### Supplemental Material

**Tables S1, Figures S1 to S4 (Spectrum02716-23-s0001.docx).** Supplemental Tables and Figures.

**Dataset S1 to S3. (Spectrum02716-23-s0002.xlsx).** Dataset to support results.

## Open Peer Review

**PEER REVIEW HISTORY (review-history.pdf).** An accounting of the reviewer comments and feedback.

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
