## [Reviewer comments · Microbiology Spectrum]

Microbiology Spectrum

Development and evaluation of the automated multipurpose molecular testing system PCRpack for high-throughput SARS-CoV-2 testing

Yasufumi Matsumura, Tomofumi Nakazaki, Kanako Kitamori, Eiki Kure, Koh Shinohara, Yasuhiro Tsuchido, Satomi Yukawa, Taro Noguchi, Masaki Yamamoto, and Miki Nagao

Corresponding Author(s): Yasufumi Matsumura, Kyoto Daigaku

Review Timeline:

Submission Date:	July 1, 2023
Editorial Decision:	August 18, 2023
Revision Received:	September 17, 2023
Accepted:	October 3, 2023

Editor: Cecilia Thompson

Reviewer(s): Disclosure of reviewer identity is with reference to reviewer comments included in decision letter(s). The following individuals involved in review of your submission have agreed to reveal their identity: Bobby G Warren (Reviewer #2)

Transaction Report:

DOI: <https://doi.org/10.1128/spectrum.02716-23>

August 18, 2023

Dr. Yasufumi Matsumura
Kyoto Daigaku
Department of Clinical Laboratory Medicine
54 Shogoin-Kawahara-cho, Sakyo-ku
Kyoto 6068507
Japan

Re: Spectrum02716-23 (Development and evaluation of the automated multipurpose molecular testing system PCRpack for high-throughput SARS-CoV-2 testing)

Dear Dr. Yasufumi Matsumura:

Link Not Available

Sincerely,

Cecilia Thompson

Journals Department
Reviewer comments:

Reviewer #1 (Comments for the Author):

In this study, Matsumura et al. successfully developed an automated PCR testing system named PCRpack, designed for the diagnosis of COVID-19. The PCRpack system combines a liquid handling instrument and a real-time PCR instrument, enabling high-throughput testing and facilitating ease of use, ensuring biosafety compliance. The results suggest that this assay holds promise in its applicability to detect SARS-CoV-2. While the PCRpack system offers several advantageous features for COVID-19 diagnosis, there are also specific issues that warrant attention and should be addressed, as indicated below.

(1) It is worth noting that the PCRpack utilized the TRexGene SARS-CoV-2 Detection Kit, which includes the internal control (FAM dye) as well as N1 (Cy5) and N2 (ROX) targets, following the manufacturer's instructions. The incorporation of the internal control is crucial in improving the accuracy of PCRpack results, as it helps identify potential false-negative samples. It is possible that some of the 15 samples that showed negative results may have given negative results under the internal control.

(2) In the Materials and Methods section, the description of all PCR conditions is unclear, and it is essential to provide this information to ensure the reproducibility of the study.

(3) The term "PCR pack" is repeated multiple times throughout the manuscript, and it appears that the space between "PCR" and "pack" may not be necessary, leading to inconsistency in the usage of the term.

Reviewer #2 (Comments for the Author):

Summary of key findings:

The manuscript describes the development and evaluation of an automated molecular testing system called PCRpack for high-throughput testing of the SARS-CoV-2 virus. The system is designed to handle the processing of a large number of samples efficiently and accurately. It includes a liquid handling instrument enclosed in a negative pressure clean booth, a laboratory information management system (LIS), a real-time PCR instrument, and other necessary equipment and documents. The performance of PCRpack was evaluated in terms of liquid handling accuracy, analytical sensitivity, clinical diagnostic performance, and testing capacity.

Liquid Handling Performance: The accuracy and precision of liquid handling by the PCRpack system were evaluated. The accuracy ranged from -2.24% to 0.73%, and the precision ranged from 0.83% to 4.52% for different volumes of reagents and samples. The system demonstrated accurate and reliable liquid handling capabilities.

Analytical Sensitivity: The limit of detection (LOD) of the PCRpack system was determined to be 1,000 copies/mL for both saliva and nasal swab samples. However, it is notable that the manual gold standard process had a LOD of 391.

Clinical Diagnostic Performance: The PCRpack system's clinical diagnostic performance was assessed using 3,965 respiratory samples. The system showed high concordance with a reference extraction-based RT-PCR assay, with positive percent agreements ranging from 95.6% to 99.0% and negative percent agreement of 100% for different specimen types.

Turnaround Time and Testing Capacity: The PCRpack system demonstrated a shorter turnaround time and a higher testing capacity compared to manual methods. The system reduced both the total elapsed time and hands-on time required for processing samples. However, it could be argued that the time difference was not substantial: "The maximum numbers of samples that could be tested within an 8-hour-day shift of one operator were 564 samples (6 batches) for PCRpack and 376 samples (4 batches) for the manual method."

Major concerns:

- The largest concern is the lack of a cost analysis. While the manuscript mentions low running costs per sample (\$3-10), it does not provide a comprehensive cost analysis that includes factors like initial investment, maintenance, and operational costs over time. These costs could vary in different settings and regions making a cost analysis difficult, however, it's a critical part of assessing the need or impact of this device, particularly when combined with a total output difference between manual and the PCRpack in an 8-hour shift was <200 samples, or a 150% increase in capacity.
- The amount of skill and training needed to use this device is also missing. The foundational argument is that larger volumes of tests are needed and there are skill staffing shortages. We can see the benefit the device brings to the former, but what about the latter?

Minor concerns:

- As mentioned above, the capacity difference seems smaller than one would expect from an automated system. I would recommend the authors explaining the rate limiting step or device - liquid handling, only 1 rt-pcr machine, etc. and how this could be increased.
- The differences in limits of detection should be characterized more and contextualized in the clinical relevance. Is this difference going to cause enough false negatives that would be concerning? Ways to attack this would be 1) reviewing retroactive data on CTs that would correlate with gene copies that would be reported as a false negative with the PCR pack or 2) reviewing the literature or clinical data to determine if CTs in this very high range generally have better clinical outcomes or not.
- This device would likely be limited to medical centers and not emergency field centers due to the PCRpack system requiring two independent power supplies of specific voltage and current specifications. This could also limit its usability in other settings with unstable power supply or inadequate power infrastructure.

Notes:

Abstract:

- Line 32:

o Lower limit of detection, correct? Clarify.

o Specify copies of what? Assumed target gene but should be clarified.

- Line 35: If possible, relay the comparison group's turnaround time per X samples and X samples per day. Since there is some discrepancy in positive test agreement (96.6%), the time and volume difference, if any, should be emphasized.

Importance:

- Data on the higher demand for testing and staff shortages should be added and the appropriate citations included. Since this is the foundational argument for the need of a faster and larger throughput process, the issue should be emphasized and characterized in depth.

Intro:

- Same comment here as from the importance - data on the need for this system. Establish the problem clearly before presenting the potential solution.

Methods:

- For clarity in the abstract and introduction, I would highlight that the PCRpack you've created utilizes a previously created and validated RT-PCR assay and emphasize to readers the real item being studied is the liquid handling.
- Characterize the hospital and geographic location to give context to the testing center. For one example: was this a large overburdened facility, as mentioned as an issue in the intro, that needed a faster/larger volume process? As a result, could some of the gold standard samples be influenced by worker fatigue or stress?
- Was the 'one skilled operator' the same person each time? If so, I would list this as a small limitation.
o Is there any data on user ability? Does this device require significant training and skill? Or could an average tech do this?
- Were efficacies between sample type analyses completed? This would allow us to determine if the liquid handler struggles with samples of different consistencies. E.g. mucus-heavy samples

Results:

- "Development of PCRpack" should be moved to the methods section.
- Was the LOD determined for the manual process? Or is there data you could cite to compare?
- The negative agreement of 100% is excellent! With liquid handling, contamination is always a large concern.

Discussion:

- Line 138: Saving labor and costs is mentioned but a full cost analysis was completed for this study. These data should be provided if available. The authors mention a running cost per sample but do not provide an upfront cost for the instruments. A cost comparison can not be completed without a rough estimate of the upfront cost as this is likely the most expensive portion and could limit many facilities from being able to use this device. If the foundational argument is that there is a high demand for testing and a lot of staff shortages, money may already be a limiting factor.
- Line 194: The LOD of the manual process should also be included in the results alongside the PCRpack's LOD. This should also be discussed more in this section. Do the authors believe the discordance in positive samples and the differences in LOD impact the overall benefit the PCRpack could provide? Is it possible to retrospectively review COVID-19 tests with the manual method for CTs that correlate to values that would be below the 1,000 LOD? Are there any data to suggest any clinical differences in patients whose test have high CTs? This would allow us to get a better idea of how many tests would have been false negatives and if those patients would likely need or not need acute care.
- The limitation section should be expanded to include some, if not all, of the following:
o Validation of the PCRpack system using a specific molecular assay (TRexGene SARS-CoV-2 Detection Kit). This might limit the generalizability of the findings to other assays or pathogens.
o While the PCRpack system reduces hands-on time and minimizes operator interventions, it still relies on skilled operators for its proper functioning. Operator expertise, training, and experience can influence the system's performance and results.
The alternative would be to comment on the device's ease of use
o While the manuscript mentions low running costs per sample, it might not provide a comprehensive cost analysis that includes factors like initial investment, maintenance, and operational costs over time. These costs could vary in different settings and regions.
o The manuscript emphasizes the customizability of the system, but the ease of customization might vary depending on the technical expertise of the laboratory staff and the complexity of modifications required.

Reviewer #3 (Comments for the Author):

The authors present a very thorough and well-written study, validating an all-in-one system to save important resources.

The study is conducted as necessary to provide robust validation of the instrument.

I only have a couple of very minor comments:

I note in the methods the 'saliva' sample is actually more like an oral swab. Can more information be provided on its collection as at present it is not clear whether this is truly a saliva sample or indeed an oral swab. It would be important to note the collection instructions so this can be evaluated by the reader. It would also be important to provide more detail on the nasal swab - what type of nasal swab (ie, AN or MT) and the collection instructions to evaluate how well a sample may or may not have been collected.

Would the system also be suitable in a mobile testing manner (ie, a lab in a van?)

Staff Comments:

Preparing Revision Guidelines

Please return the manuscript within 60 days; if you cannot complete the modification within this time period, please contact me. If you do not wish to modify the manuscript and prefer to submit it to another journal, please notify me of your decision immediately so that the manuscript may be formally withdrawn from consideration by Microbiology Spectrum.

**Replies to the comments raised by the Editor**

Thank you very much for your review of the manuscript. We are grateful to the reviewers for
suggesting important modifications needed for the manuscript. We have thoughtfully taken them
into account. The responses to the reviewers' suggestions are given point by point in the following
pages.

To follow the journal's recommendations, we added CRediT taxonomy CRediT author contribution
statement as follows:

“Yasufumi Matsumura: Conceptualization, Data curation, Formal Analysis, Funding acquisition,
Investigation, Methodology, Project administration, Resources, Supervision, Validation,
Visualization, Writing - original draft, Writing - review & editing Tomofumi Nakazaki:
Conceptualization, Data curation, Funding acquisition, Methodology, Resources, Investigation,
Software, Validation, Writing - review & editing Kanako Kitamori: Funding acquisition,
Methodology, Investigation, Software, Writing - review & editing Eiki Kure: Conceptualization,
Investigation, Validation, Writing - review & editing Koh Shinohara: Investigation, Writing - review
& editing Yasuhiro Tsuchido: Investigation, Writing - review & editing Satomi Yukawa:
Investigation, Writing - review & editing Taro Noguchi: Investigation, Writing - review & editing
Masaki Yamamoto: Investigation, Data curation, Resources, Writing - review & editing Miki
Nagao: Funding acquisition, Resources, Supervision, Writing - review & editing.” (page 16, line
387, in the revised manuscript)

We added Dataset S1 to S3 and “Data availability” paragraph to fulfill the Open Data Policy of the
ASM Editorial Policies.

“Data availability

All data used in this study are available within the manuscript and supplemental materials.” (page

15, line 380)

“Raw data are available in Dataset S1.” (Figure S2 legend)

“Raw data are available in Dataset S2.” (Figure 3 legend)

We used a professional English editing service (American Journal Experts) to proofread the revised
manuscript. Following changes were made but the meanings of the sentences or words were
unchanged.

“The developed PCRpack system shows high liquid handling and clinical diagnostic performance
and is a promising testing system for increasing the SARS-CoV-2 testing capacity and ~~for~~ testing
future emerging pathogens.” (page2, line 39; “for” was removed.)

“We developed an automated multipurpose molecular testing system, named PCRpack, for the rapid,
high-throughput testing of infectious pathogens, including SARS-CoV-2. The system is provided in
an all-in-one package, including a liquid handling instrument, a laboratory information management
system, and other materials needed for testing operation; is highly customizable; and is easily
implemented.” (page2, line 45; “others” was replaced by “other materials” and comma/semicolons
were added.)

“These features contribute to the rapid implementation of the high-performance and
high-throughput molecular testing environment at any phase of the pandemic caused by
SARS-CoV-2 or future emerging pathogens.” (page 2, line 52; “due to” was replaced by “caused
by” and “and” was replaced by “or”.)

“PCRpack used a TRexGene® SARS-CoV-2 Detection Kit (Toyobo, Osaka, Japan) for direct
RT-PCR testing, which bypasses a nucleic acid purification step.” (page 5, line 97)

“A total of 3965 respiratory samples (saliva, n=2198; nasopharyngeal swabs, n=1177, nasal swabs,
n=590) were included in the analysis.” (page 5, line 107; “for” was replaced by “in”.)

“The LIS of PCRpack, SimpPCR, can integrate different instrumentation and provide accurate

sample tracking, quality control, result interpretation, and reporting, contributing to accurate testing
and shortening turn-around times.” (page 7, line 153; commas were added after “PCRpack”.)
“The Roche Cobas 8800 and Hologic Panther Fusion systems have higher capacities, but detailed
data in real-life settings are lacking for these systems.” (page 10, line 222; “such” appeared before
“detailed data” was deleted.)
“8-hour-day shift” was changed to “8-hour shift”. (page 2, line 36 and elsewhere)
“Thermo Fischer Scientific” was corrected to “Thermo Fisher Scientific”. (page 13, line 305 and
elsewhere)

**Replies to the comments raised by Reviewer #1**

In this study, Matsumura et al. successfully developed an automated PCR testing system named
PCRpack, designed for the diagnosis of COVID-19. The PCRpack system combines a liquid
handling instrument and a real-time PCR instrument, enabling high-throughput testing and
facilitating ease of use, ensuring biosafety compliance. The results suggest that this assay holds
promise in its applicability to detect SARS-CoV-2. While the PCRpack system offers several
advantageous features for COVID-19 diagnosis, there are also specific issues that warrant attention
and should be addressed, as indicated below.

**Response.** Thank you very much for your review of the manuscript and for providing valuable
comments to improve our manuscript.

(1) It is worth noting that the PCRpack utilized the TRexGene SARS-CoV-2 Detection Kit, which
includes the internal control (FAM dye) as well as N1 (Cy5) and N2 (ROX) targets, following the
manufacturer's instructions. The incorporation of the internal control is crucial in improving the
accuracy of PCRpack results, as it helps identify potential false-negative samples. It is possible that
some of the 15 samples that showed negative results may have given negative results under the
internal control.

**Response.** We agree that the inclusion of an internal control is very important, and this is a strength
of PCRpack. Following your comments, we added explanations related to the internal control as
follows:

“No samples were classified as invalid, including 15 discordant samples (saliva, n=3;
nasopharyngeal swab, n=11; nasal swab, n=1) that showed negative results by PCRpack (with a
positive internal control) and positive results by the reference assay.” (page 4, line 112, in the

revised manuscript)

“Fluorescent signals of the SARS-CoV-2 N1 (Cy5 dye) and N2 (ROX) targets and the internal
control (FAM) were detected on the QIAquant® 96 5plex (Qiagen, Hilden, Germany) real-time
PCR instrument.” (page 12, line 289)

(2) In the Materials and Methods section, the description of all PCR conditions is unclear, and it is
essential to provide this information to ensure the reproducibility of the study.

**Response.** We agree with your comment. We added descriptions of all PCR conditions as follows:

“An in vitro diagnostics TRexGene® SARS-CoV-2 Detection Kit was used for direct RT-PCR
testing according to the manufacturer’s instructions. First, 8 µl of raw samples was mixed with 3 µl
of sample treatment reagents, followed by heating at 95 °C for 5 minutes and the addition of 40 µl
of RT-PCR reagents. Fluorescent signals of the SARS-CoV-2 N1 (Cy5 dye) and N2 (ROX) targets
and the internal control (FAM) were detected on the QIAquant® 96 5plex (Qiagen, Hilden,
Germany) real-time PCR instrument. Thermal cycling was performed as follows: 5 minutes at 42 °C
and 10 seconds at 95 °C, followed by 45 cycles of 5 seconds at 95 °C and 30 seconds at 60 °C. Ct
values were calculated by using the automatic determination of the fluorescence threshold function.
Results were interpreted as positive when Ct values were < 40 for either the N1 or N2 genes and
negative when Ct values were > 40 for both N1 and N2 genes and the Ct value of the internal
control was <40. All other results were regarded as invalid. For quality control, positive and
negative controls from the AccuPlex™ SARS-CoV-2 Reference Material Kit (SeraCare, Milford,
MA, USA) were used for each batch.” (page 12, line 286))

“The N2 assay developed by the National Institute of Infectious Disease in Japan (NIID)
(21) was used as the reference standard. RNA was extracted from 200 µL respiratory samples using
the MagNA Pure 96 DNA and Viral NA Small Volume extraction kit and a MagNA Pure 96

Instrument (Roche, Basel, Switzerland) and was eluted in a final volume of 50 μ L. RT-PCR was
performed using 5 μ L of extracted RNA, 0.5 pM of NIID_2019-nCOV_N_F2 forward primer
(AAATTTTGGGGACCAGGAAC), 0.7 pM of NIID_2019-nCOV_N_R2 reverse primer
(TGGCAGCTGTGTAGGTCAAC), 0.2 pM of NIID_2019-nCOV_N_P2 probe
(FAM-ATGTCGCGCATTGGCATGGA-BHQ1), and TaqPath™ 1-Step RT-qPCR Master Mix, CG
(Thermo Fisher Scientific) in a 20 μ L reaction volume on a LightCycler® 480 System II (Roche,
Basel, Switzerland). Thermal cycling was performed as follows: 2 minutes at 25 °C, 15 minutes at
50 °C and 2 minutes at 95 °C, followed by 45 cycles of 3 seconds at 95 °C and 30 seconds at 60 °C.
The cycle threshold (Ct) values were determined by the second derivative maximum method. The
results were interpreted as positive when the Ct value of the N2 gene was < 40 and as negative when
it was \geq 40. The samples for which the results were discordant between PCRpack and reference
assay were tested using the in vitro diagnostics Xpert® Xpress SARS-CoV-2 assay according to the
manufacturer’s instructions.” (page 13, line 343)
The sentence describing invalid results of the N2 reference assay was deleted because the invalid
criteria do not exist.

(3) The term "PCR pack" is repeated multiple times throughout the manuscript, and it appears that
the space between "PCR" and "pack" may not be necessary, leading to inconsistency in the usage of
the term.

**Response.** As you observed, “PCRpack” is a proper noun and should not include a space between
two words. We double-checked the manuscript thoroughly and confirmed that the misspelled term

“PCR pack” was changed.

**Replies to the comments raised by Reviewer #2**

Thank you very much for your review of the manuscript and for providing valuable comments to
improve our manuscript.

Major concerns:

- The largest concern is the lack of a cost analysis. While the manuscript mentions low running costs
139 per sample (\$3-10), it does not provide a comprehensive cost analysis that includes factors like
initial investment, maintenance, and operational costs over time. These costs could vary in different
settings and regions making a cost analysis difficult, however, it's a critical part of assessing the
need or impact of this device, particularly when combined with a total output difference between
manual and the PCRpack in an 8-hour shift was <200 samples, or a 150% increase in capacity.

**Response.** We agree with your comment. It is difficult to determine the initial investment and
maintenance costs, as these are only available on application by a local distributor. This is true for
other fully automated commercial systems, and this information is usually not available in public
data sources, including scientific papers. The maintenance of PCRpack does not require component
replacement at regular intervals, but an annual maintenance inspection is recommended. We
re-calculated the running costs to include consumables for testing and labor costs based on the
current situations, as the best possible measure. Consumables costs included reagents, controls,
pipette tips, plates and other plasticware needed for tests. With 94 samples per batch, the
consumables cost including reagents, controls, pipette tips, plates and other plasticware for one
sample was estimated as \$12.5 for PCRpack and \$11.9 for the manual method. The labor costs
needed for testing within an 8-hour shift were \$64 for PCRpack and \$140 for the manual method, as
calculated by the total hands-on time within an 8-hour shift multiplied by labor cost per hour (\$25).
The total running costs (reagents/consumables plus labor) per sample were calculated as \$12.6 for
PCRpack and \$12.3 for the manual method.

The total output difference between PCRpack and the manual method may not be large
enough from the view of cost effectiveness because PCRpack needs an additional \$0.3/sample for
testing plus initial and maintenance costs. A smaller number of tests makes the depreciation of the
initial and maintenance costs per sample even larger. However, the main strengths of PCRpack are
the accuracy and quality of testing while not requiring a longer working time of highly skilled
technicians. The running cost of PCRpack (\$12.6) is sufficiently lower than that of other fully
automated commercial systems that provide similarly accurate and high-quality testing (\$40–60 per
sample). This information, including the limitations for cost analysis, were incorporated into the
manuscript as follows:

**“Test time and cost**

(omitted)

With 94 samples per batch, the cost for one sample was \$12.6 for PCRpack and \$12.3 for the
manual method: \$12.5 and \$11.9 for consumables and \$64 (2 h 33 m hands-on time per 564
samples) and \$140 (5 h 35 m per 376 samples) for labor, respectively (Dataset S3).” (page 6, line
118 and 123)

**“Test time and cost**

(omitted)

The consumables cost of PCRpack may be lower than that of other fully automated platforms
(approximately \$13 vs. \$40–60 in Japan). Compared with the manual method, PCRpack requires an
additional running cost of \$0.3/sample plus initial and maintenance costs. The maintenance of
PCRpack does not require component replacement at regular intervals, but an annual maintenance
inspection is recommended. The cost effectiveness and merits of PCRpack depend on individual

situations, including requirements for testing capacity and availability of skilled staff. A larger
number of tests may increase the risk for error in the manual method, giving the accurate automated
testing by PCRpack an advantage.” (page 9, line 216 and page 10, line 228)

**“Test time measurement and cost estimate**

The time needed to perform each test step for the testing of 94 samples was measured in
triplicate for the PCRpack and manual methods. Trials were performed by the same skilled operator
on different days. In the manual method, the same RT-PCR testing performed by the PCRpack
system was performed without the Biomek i5 instrument, but SimpPCR was used for sample
management and RT-PCR result interpretation (Figure S1). The turnaround time and timeline of
each batch within an 8-hour-day shift were calculated. Running costs for the PCRpack and manual
methods were calculated from the costs for consumables (reagents and plasticware) and labor (the
total hands-on time within an 8-hour shift multiplied by labor costs per hour (\$25). Costs were
estimated at an exchange rate of \$1 = 120 yen.” (page 15, line 359)

- The amount of skill and training needed to use this device is also missing. The foundational
argument is that larger volumes of tests are needed and there are skill staffing shortages. We can see
the benefit the device brings to the former, but what about the latter?

**Response.** As you observed, the comparison of skill level and time needed for training are very
important points to characterize PCRpack. In our laboratory at Kyoto University, the standard
training period for PCRpack requires at least 3 days, while the manual method requires 10 days,
although it depends on the aptitude and baseline skill of the staff member. The skills and training

needed to operate PCRpack were clarified as follows:

“The operation of PCRpack is easier than the manual method due to the automation of testing steps.

In our laboratory, less time was needed for staff training for PCRpack (3 vs. 10 days) than for the

manual method. This feature, combined with the shorter hands-on time, can be strengths of

PCRpack in cases of skilled staff shortages.” (page 7, line 163)

“A PCRpack operator needs basic laboratory skills (pipetting, centrifuge, and vortex). The 3-day

training program for a technical assistant includes proficiency in the standard operating procedure;

operation of the instruments (liquid handler and PCR instrument) and LIS; and knowledge of

biosafety level 2, assay interpretation, reporting, and quality assurance.” (page 12, line 281)

Minor concerns:

- As mentioned above, the capacity difference seems smaller than one would expect from an

automated system. I would recommend the authors explaining the rate limiting step or device -

liquid handling, only 1 rt-pcr machine, etc. and how this could be increased.

**Response.** As you observed, the capacity difference between PCRpack and fully automated

commercial systems is small. The PCRpack system may require a slightly larger amount of labor.

The main advantage of PCRpack over other automated systems is the flexibility to utilize various

reagents/labware, which allows for the robustness to continue testing during supply shortages and

the capability to change assays for other pathogens. To scale up PCRpack, an additional PCR

instrument is a cost-effective method. Following your recommendation, we added discussion on

increasing the capacity of PCRpack as follows.

“The testing capacity of PCRpack is limited by the PCR step rather than the liquid handling step

(Figure S3). Adding another PCR instrument can enhance the testing capacity to 752 samples (8

batches) within an 8-hour shift (Figure S4).” (page 10, line 225)

- The differences in limits of detection should be characterized more and contextualized in the
clinical relevance. Is this difference going to cause enough false negatives that would be
concerning? Ways to attack this would be 1) reviewing retroactive data on CTs that would correlate
with gene copies that would be reported as a false negative with the PCR pack or 2) reviewing the
literature or clinical data to determine if CTs in this very high range generally have better clinical
outcomes or not.

**Response.** The false negatives observed in this study were due to the lower analytical sensitivity of
the TRexGene assay compared with the reference assay (NIID N2 assay with nucleic acid
extraction). As you pointed out in another comment, this manuscript evaluates PCRpack's liquid
handling and integrated system and does not evaluate previously validated RT-PCR assays.
Therefore, we think that further analysis of false-negative samples may be beyond the scope of the
manuscript. We apologize, but your suggestions were difficult to answer. For the first question, the
Ct values of false-negative samples were very high, and the majority of samples were outside of the
gene quantification range of >50 copies/reaction (Ct value of <34). We also think that it is difficult
to determine the clinical relevance of low copy number samples from literature reviews because
assays, specimens, and patients greatly differ among studies. Unfortunately, it was not possible to
obtain adequate clinical information for false-negative samples in this study. This was added to the
study limitations.

“Third, we could not assess the clinical significance of false-negative samples due to a lack of
clinical information.” (page 10, line 242)

- This device would likely be limited to medical centers and not emergency field centers due to the
PCRpack system requiring two independent power supplies of specific voltage and current
specifications. This could also limit its usability in other settings with unstable power supply or
inadequate power infrastructure.

**Response.** We agree with your point. We added this information as follows:

“The need for two independent power supplies may restrict its utility in emergency field centers
with an inadequate or unstable power supply.” (page 8, line 168)

Notes:

Abstract:

- Line 32:

o Lower limit of detection, correct? Clarify.

**Response.** We amended the phrase to “lower limit of detection”.

“Analytical sensitivity analysis showed a lower limit of detection of 1,000 genome copies/mL of
sample.” (page 2, line 31)

o Specify copies of what? Assumed target gene but should be clarified.

**Response.** We clarified that this is the genome copy number. The viral genome concentrations used
in the analytical sensitivity analysis were given by the manufacturer of the reference standard
(ATCC).

“Analytical sensitivity analysis showed a lower limit of detection of 1,000 genome copies/mL of
sample.” (page 2, line 31)

- Line 35: If possible, relay the comparison group's turnaround time per X samples and X samples
271 per day. Since there is some discrepancy in positive test agreement (96.6%), the time and volume
difference, if any, should be emphasized.

**Response.** We added data for the manual method for comparison as follows:

“The average turnaround times per 94 samples and the maximum numbers of tests within an 8-hour
shift of one operator were 2 hours 28 minutes vs. 2 hours 1 minute and 564 vs. 376 samples for

PCRpack and the manual method, respectively.” (page 2, line 35)

Importance:

- Data on the higher demand for testing and staff shortages should be added and the appropriate
citations included. Since this is the foundational argument for the need of a faster and larger
throughput process, the issue should be emphasized and characterized in depth.

**Response.** We elaborated the importance section to emphasize the higher demand for testing and
staff shortages. However, according to the journal’s formatting guidelines, in the importance section,
citations are not allowed, and the word count is restricted to 150 words.

“Accurate and fast molecular testing is important for the diagnosis and control of COVID-19.

During patient surges in the COVID-19 pandemic, laboratories were challenged by a higher demand
for molecular testing under skilled staff shortages.” (page 2, line 43)

Intro:

- Same comment here as from the importance - data on the need for this system. Establish the
problem clearly before presenting the potential solution.

**Response.** We added sentences that describe the high demand for testing in Japan and staff
shortages in the Introduction as follows:

“Japan has experienced several COVID-19 waves (3), and there has been both a continuous demand
and surges in demand for molecular tests. The average number of people who underwent RT-PCR
tests per day before the first wave (April 2020) was 569, but it expanded to 19,923 during the third
wave (August 2020) and 161,992 in 2022, according to the Ministry of Health, Labour and Welfare
of Japan (https://www.mhlw.go.jp/stf/covid-19/kokunainohasseijoukyou_00006.html). An up to
five-fold increase in the number of tests was observed in the 6th wave that started in January 2022.

At the same time, in this COVID-19 pandemic, we have been challenged by global supply shortages

and the need for skilled laboratory professionals (4, 5). Molecular tests, especially
laboratory-developed or in-house assays (e.g., the World Health Organization-accredited Corman's
assay) (6), require specialized skills (7). Adding new staff may be difficult because hiring and
training staff members while responding to the high demand for tests requires substantial effort, and
training for molecular diagnostics involves a robust education curriculum (8, 9)." (page 4, line 65)

Methods:

- For clarity in the abstract and introduction, I would highlight that the PCRpack you've created
utilizes a previously created and validated RT-PCR assay and emphasize to readers the real item
being studied is the liquid handling.

**Response.** To clarify that PCRpack used a previously validated in vitro diagnostic assay and that the
manuscript evaluated the system performance, including liquid handling, rather than the assay itself,
we added the following sentences to the Abstract and the Introduction. To reduce word count, we
deleted the acronym expansions of SARS-CoV-2 and COVID-19 from the manuscript, as they have
been defined as proper nouns by WHO and ICTV, respectively.

"An in vitro diagnostic assay was employed to detect SARS-CoV-2." (page 2, line 27)

"PCRpack employs an in vitro diagnostic assay to detect SARS-CoV-2. In this study, we aimed to
evaluate the PCRpack system and determine its liquid handling performance, analytical sensitivity,
clinical diagnostic performance, and testing capacity for the molecular detection of SARS-CoV-2."
(page 5, line 90)

- Characterize the hospital and geographic location to give context to the testing center. For one
example: was this a large overburdened facility, as mentioned as an issue in the intro, that needed a
faster/larger volume process? As a result, could some of the gold standard samples been influenced
my worker fatigue or stress?

**Response.** We added the location and a brief description of the hospital. The evaluations, including
the clinical study, were conducted after prospective storage of the primary tests. Therefore, the study
was not influenced by worker fatigue or stress. This fact was also added.

“Kyoto University Hospital is a 1,141-bed tertiary academic center located in Kyoto, Japan which
has a population of \approx 1 million.” (page 13, line 324)

“The samples that showed an adequate remaining volume for the study were eligible and were
prospectively stored at -80°C until tested by PCRpack and the reference assay. Therefore, clinical
diagnostic performance study was not influenced by overburdened settings during surges for tests.”
(page 14, line 338)

- Was the 'one skilled operator' the same person each time? If so, I would list this as a small
limitation.

**Response.** Yes, the same person was involved in the measurements. This was clarified by changing
“one skilled operator” to “the same skilled operator”. Using the data from only one operator was
added to the limitations.

“First, liquid handling performance and test time measurements were performed by only one
operator, and differences and/or variabilities among operators could not be evaluated.”

o Is there any data on user ability? Does this device require significant training and skill? Or could
an average tech do this?

**Response.** We added the skills and training needed to operate the PCRpack as follows:

“A PCRpack operator needs basic laboratory skills (pipetting, centrifuge, and vortex). The 3-day
training program for a technical assistant includes proficiency in the standard operating procedure;
operation of the instruments (liquid handler and PCR instrument) and LIS; and knowledge of
biosafety level 2, assay interpretation, reporting, and quality assurance.” (page 10, line 237)

- Were efficacies between sample type analyses completed? This would allow us to determine if the
liquid handler struggles with samples of different consistencies. E.g. mucus-heavy samples

**Response.** In the liquid handling performance and clinical diagnostic performance studies, we used
clinical saliva samples, which might be more viscous than nasal or nasopharyngeal swabs. It is
important to use nonviscous samples to ensure liquid handling performance. Therefore, saliva
samples were collected using a swab and submerged in viral transport media, reducing the viscosity.
Detailed saliva collection procedures were added to the Methods (partly reflecting other reviewers'
comments). All samples were centrifuged before testing; this contributed to avoiding the
mucus-heavy samples. In the clinical diagnostic performance study, there were no apparent
differences among sample types (Table 2). This was added to the discussion.

“Liquid handlers have a potential risk of errors with viscous samples (such as saliva), but there were
no differences in diagnostic performance among the different sample types (Table 2). To avoid
dispensing problems, we utilized a swab-based saliva sampling method that traps mucous secretions
and centrifuged all samples to avoid dispensing mucus-heavy portions.” (page 9, line 207)

“Saliva was collected by placing a proprietary swab (Sysmex, Kobe, Japan) that can absorb 1 mL of
saliva in mouth without rubbing for at least one minute in compliant with manufacturer’s
instructions. Test examinees were asked not to eat or drink 30 minutes before collection of saliva.”
(page 13, line 325)

“After collection procedures for saliva, nasal and nasopharyngeal swabs, they were submerged in 3
371 mL of viral transport media and their shafts were broken to leave the swabs in the media.” (page 14,
line 336)

Results:

- "Development of PCRpack" should be moved to the methods section.

**Response.** This information has been moved to the methods section. In this section, system
components and workflows were described. To help readers understand important characteristics of
the system, we added minimal descriptions to the Introduction.

“PCRpack includes a customized liquid handling instrument (Biomek i5, Beckman Coulter, Tokyo,
Japan), a laboratory information management system (LIS) (SimpPCR, Nippon Control System,
Yokohama, Japan), a real-time PCR instrument, other equipment needed for the molecular detection
of SARS-CoV-2, and documents or templates for testing and laboratory management (e.g., standard
operating procedure; Figure 1). PCRpack employs an in vitro diagnostic assay to detect
SARS-CoV-2. In this study, we aimed to evaluate the PCRpack system and determine its liquid
handling performance, analytical sensitivity, clinical diagnostic performance, and testing capacity
for the molecular detection of SARS-CoV-2. In this study, we aimed to evaluate the PCRpack
system and determine its liquid handling performance, analytical sensitivity, clinical diagnostic
performance, and testing capacity for the molecular detection of SARS-CoV-2.” (page 5, line 85)

- Was the LOD determined for the manual process? Or is there data you could cite to compare?

**Response.** The LOD for the TRexGene assay by the manual method was determined in our previous
study (Matsumura et al. J Clin Virol Plus. 2023 Feb;3(1):100138). This was added to the “Analytical
sensitivity” section in the Discussion.

“The defined LODs of the PCRpack system (1,000 copies/mL) were the same as the results from
our previous study performed with a SARS-CoV-2 Detection Kit -Multi- (the former product name
of TRexGene) by manual handling (17).” (page 9, line 194)

- The negative agreement of 100% is excellent! With liquid handling, contamination is always a
large concern.

**Response.** Thank you for your comment.

Discussion:

- Line 138: Saving labor and costs is mentioned but a full cost analysis was completed for this study.

These data should be provided if available. The authors mention a running cost per sample but do
not provide an upfront cost for the instruments. A cost comparison can not be completed without a
rough estimate of the upfront cost as this is likely the most expensive portion and could limit many
facilities from being able to use this device. If the foundational argument is that there is a high
demand for testing and a lot of staff shortages, money may already be a limiting factor.

**Response.** We agree that money is the limiting factor for those considering the introduction of
automated systems, and a comparison of full costs is needed. However, as we responded to your
major concern (presented above), it is difficult to determine the initial cost of PCRpack and other
fully automated commercial systems. We think the comparison needs to be done by researchers who
are searching for these types of systems. We reperformed the cost analysis to include labor costs to
make the cost analysis better. Please see the above response to your major concern.

- Line 194: The LOD of the manual process should also be included in the results alongside the
PCRpack's LOD. This should also be discussed more in this section. Do the authors believe the
discordance in positive samples and the differences in LOD impact the overall benefit the PCRpack
could provide? Is it possible to retrospectively review COVID-19 tests with the manual method for
CTs that correlate to values that would be below the 1,000 LOD? Are there any data to suggest any
clinical differences in patients whose test have high CTs? This would allow us to get a better idea of
how many tests would have been false negatives and if those patients would likely need or not need
acute care.

**Response.** The false negatives observed in this study were due to the lower analytical sensitivity of
the TRexGene assay compared with the reference assay (NIID N2 assay, with nucleic acid

extraction). As you observed in another comment, this manuscript evaluates PCRpack’s liquid
handling and integrated system and does not evaluate previously validated RT-PCR assays.
Therefore, we think that further analysis of false-negative samples may be beyond the scope of the
manuscript. We apologize, but the suggestions kindly provided in your comment were both difficult
to answer. For the first question, the Ct values of false-negative samples were very high, and the
majority of samples were outside of the gene quantification range of >50 copies/reaction (Ct value
of <34). We also think that it is difficult to determine the clinical relevance of low copy number
samples from literature reviews because assays, specimens, and patients greatly differ among
studies. Unfortunately, it was not possible to obtain adequate clinical information for false-negative
samples in this study. This was added to the study limitation.

“Third, we could not assess the clinical significance of false-negative samples due to a lack of
clinical information.” (page 10, line 242)

- The limitation section should be expanded to include some, if not all, of the following:

o Validation of the PCRpack system using a specific molecular assay (TRexGene SARS-CoV-2
Detection Kit). This might limit the generalizability of the findings to other assays or pathogens.

**Response.** We would like to express our sincere gratitude to the reviewer for making great
comments/proposals to improve our manuscript. We agree that the generalizability is limited by
validation performed only with the TRexGene assay. In the original manuscript, this was described
in the limitations as a “lack of validation by different molecular assays”. To clarify that this fact is
related to limitations in generalizability, we revised the sentence as follows. The limitation of
“potential sampling bias of clinical samples” was removed because it could be included in the
revised sentence of “lack of validation by samples obtained from different clinical backgrounds”.

“This study has several limitations. First, liquid handling performance and test time measurements
were performed by only one operator, and differences and/or variabilities among operators could not

be evaluated. Second, the study lacked validation by multiple investigators or in multiple locations,
different molecular assays, and samples obtained from different clinical backgrounds.” (page 10,
line 237)

o

While the PCRpack system reduces hands-on time and minimizes operator interventions, it still
relies on skilled operators for its proper functioning. Operator expertise, training, and experience
can influence the system's performance and results.

**Response.** This limitation is added as follows.

“Fourth, while PCRpack was designed to reduce hands-on time and operator interventions, it still
relies on skilled operators for its proper functioning. Operator expertise, training, and experience
can influence the system's performance.” (page 10, line 243)

□ The alternative would be to comment on the device's ease of use

o While the manuscript mentions low running costs per sample, it might not provide a
comprehensive cost analysis that includes factors like initial investment, maintenance, and
operational costs over time. These costs could vary in different settings and regions.

**Response.** The limitation for cost analysis was added as follows.

“Fifth, we could not perform a comprehensive analysis including initial investment, maintenance,
and operational costs over time. These costs could vary in different settings and regions.” (page 10,
line 246)

o The manuscript emphasizes the customizability of the system, but the ease of customization might
vary depending on the technical expertise of the laboratory staff and the complexity of
modifications required.

**Response.** It is true that the ease of customization might vary depending on the complexity of the
modifications needed. It is very easy to change reagent volumes and/or sample types because they
can be performed by the same operation system for PCRpack. Changes in protocols will be made by
the manufacturer and are not associated with the technical expertise of the laboratory staff. This has
been explained in the Discussion section. However, according to the level of customization, extra
validation studies by skilled laboratory staff may be needed. This was added as follows:
“In the SimpPCR operating window, the specimen types and transfer volumes of reagents and/or
samples can be modified. Changes in testing protocols (number of reagents, editing of testing steps)
are also possible upon request, but they may require validation studies by a skilled laboratory staff
according to the level of customization.” (page 7, line 140)

**Replies to the comments raised by Reviewer #3**

Thank you very much for your review of the manuscript and for providing valuable comments to
improve our manuscript.

The authors present a very thorough and well-written study, validating an all-in-one system to save
important resources.

The study is conducted as necessary to provide robust validation of the instrument.

I only have a couple of very minor comments:

I note in the methods the 'saliva' sample is actually more like an oral swab. Can more information be
provided on its collection as at present it is not clear whether this is truly a saliva sample or indeed
an oral swab. It would be important to note the collection instructions so this can be evaluated by
the reader. It would also be important to provide more detail on the nasal swab - what type of nasal
swab (ie, AN or MT) and the collection instructions to evaluate how well a sample may or may not
have been collected.

**Response.** We agree with your comment. We added detailed information for specimen collection. In
brief, saliva was collected by placing a proprietary swab for the collection of saliva into the mouth
without rubbing, and the swab was submerged into the transport media. We used an anterior nasal
swab for nasal specimen collection.

“Saliva was collected by placing a proprietary swab (Sysmex, Kobe, Japan) that can absorb 1 mL of
saliva in mouth without rubbing for at least one minute in compliant with manufacturer’s
instructions. Test examinees were asked not to eat or drink 30 minutes before collection of saliva.

Flocked swabs (Copan) were used for the collection of anterior nasal and nasopharyngeal specimens.

For nasal swab collection, a swab was inserted into the first nostril until the swab tip is no longer
visible and rotated against the wall of the nostril in a large circular path 5 times. The same swab was
used for the specimen collection of the other nostril and the same procedure was repeated. For
nasopharyngeal swab collection, a swab was into the nostril, parallel to the palate until the swab
reached a depth equal to the distance from the nostrils to the outer opening of the ear or the
examiner detected resistance. The swab was left in place for 10 seconds to absorb secretions and
then it was removed slowly while rotating it. After collection procedures for saliva, nasal and
nasopharyngeal swabs, they were submerged in 3 mL of viral transport media and their shafts were
broken to leave the swabs in the media.” (page 13, line 325)

Would the system also be suitable in a mobile testing manner (ie, a lab in a van?)

**Response.** We planned to load the system in a van to organize a mobile laboratory. The dimensions
of the system are compliant with a large size, high-height van, but we have not actually loaded the
system nor validated it. This was added to the Discussion.

“The size of the PCRpack system is compliant with a large size, high-height van (Figure 1). Future
studies are needed to optimize and validate a mobile PCRpack laboratory.” (page 8, line 167)

October 3, 2023

Dr. Yasufumi Matsumura
Kyoto Daigaku
Department of Clinical Laboratory Medicine
54 Shogoin-Kawahara-cho, Sakyo-ku
Kyoto 6068507
Japan

Re: Spectrum02716-23R1 (Development and evaluation of the automated multipurpose molecular testing system PCRpack for high-throughput SARS-CoV-2 testing)

Dear Dr. Yasufumi Matsumura:

Your manuscript has been accepted, and I am forwarding it to the ASM Journals Department for publication. You will be notified when your proofs are ready to be viewed.

Sincerely,

Cecilia Thompson
Editor, Microbiology Spectrum
